# Relationship between knowledge, attitude, and practice of COVID-19 precautionary measures and the frequency of infection among medical students at an Egyptian University

**Ghada O. Wassif●\*◎, Dina Ahmed Gamal El Din◎**

Department of Community, Environmental & Occupational Medicine, Faculty of Medicine, Ain Shams University, Cairo, Egypt

◎ These authors contributed equally to this work.
\* ghada_wassif@yahoo.com, ghadawasif@med.asu.edu.eg

**Data Availability Statement:** All relevant data are within the paper and its Supporting Information files.

## Abstract

### Background

Medical undergraduates are at high risk of COVID-19 infection. Thus, conformance to healthy practices is advised to reduce disease transmission and control the current epidemic. The present study aimed to explore the relationship of knowledge, attitude, and practice (KAP) related to COVID-19 precautionary measures with the frequency of infection among medical students at an Egyptian University.

### Methods

A comparative cross-sectional study was conducted on 404 undergraduate medical students from different grades using a web-based self-administered anonymous questionnaire.

### Results

More than one-third of medical students (37.4%) were previously infected with COVID-19, where the majority (60.5%) were diagnosed with relevant signs and symptoms. Medical students with low levels of KAP experienced higher frequencies of infection than did other students. A statistically significant negative correlation was observed between the number of previous COVID-19 infections among medical students and their knowledge and attitude scores toward COVID-19. In addition, a statistically significant positive correlation was noted among KAP scores ($P < 0.01$).

### Conclusion

Improving the knowledge, attitude, and conformance of medical students to precautionary measures toward COVID-19 may substantially reduce the risk and frequency of infection and, hence, reduce community transmission.

**Funding:** The author(s) received no specific funding for this work.

**Competing interests:** The authors have declared that no competing interests exist.

## Background

COVID-Coronavirus Disease 2019 (COVID-19) is a rapidly expanding pandemic caused by a novel human coronavirus (SARS-CoV-2) previously known as 2019-nCoV [1,2]. COVID-19 was first reported in December 2019 among patients with symptoms of viral pneumonia in Wuhan, China [3,4].

Knowledge, attitude, and practice (KAP) are vital in controlling the spread of the illness. Knowledge about the cause of the disease and its signs/symptoms and conceivable strategies for avoidance can promote the proactive application of preventive measures [5].

Hesham et al. [6] conducted a cross-sectional study on 439 undergraduate medical students (1st–6th academic years) to assess knowledge, attitude, and preventive practices related to COVID-19 using an online questionnaire. The authors found that Egyptian medical students possessed acceptable levels of knowledge, positive attitude, and good practices of preventive measures related to COVID-19.

COVID-19 has postponed the training of medical students across universities due to school closure during the lockdown. During pandemics, such as COVID-19, the healthcare system is placed under immense pressure, such that it forces authorities to recruit medical undergraduates to provide medical care to patients, which exposes them to the risk of transmitted infection [7].

Moreover, medical students represent common references for healthcare advice for family members and friends [8,9], particularly senior students (clinical stages) [10].

This study conducted an extensive literature review of articles that addressed the relationship between KAP related to COVID-19 precautionary measures and the frequency of infection among medical students and found no previous studies on this topic. Nevertheless, we found a study that explored the relationship between the KAP of healthcare workers and infection status, which was conducted by Ghonaim et al. [11]. The study recruited 206 health care workers (from the COVID-19 isolation unit of the National Liver Institute, Menoufia University, Egypt, in the period from June 10 to August 10, 2020. The results indicated that health care workers with negative SARS-CoV-specific polymerase chain reaction (PCR), serum total antibodies, and normal CT scan exhibited significant scores for knowledge and high levels of positive attitude.

The present study intends to explore if a relationship exists between the KAP of medical students related to COVID-19 precautionary measures and the frequency of infection in an endeavor to provide evidence that the conformance of medical students to COVID-19 precautionary measures can substantially reduce the incidence of infection. This objective is in line with the notion that infection may occur once due to inevitable contact with an infected family member. However, frequent infection could be prevented by practicing precautionary measures, which could decrease the burden of COVID-19 on the health care system and reduce the cost of treatment. In addition, mortalities associated with COVID-19 could be reduced effectively.

## Materials and methods

### Study participants and design

This study is a comparative cross-sectional one conducted at the Faculty of Medicine, Ain Shams University located in Abasia Square in Cairo and is one of the largest educational medical institutions in Africa and the Middle East. It was founded in 1947, which makes it the third-oldest medical school in Egypt. It has promoted numerous programs of medical care to serve society in addition to continued scientific research for local and international health. The

participants were undergraduate medical students in their first to sixth academic years who agreed to participate in the study. Students were recruited across three months, that is, from June to August 2021 via the snowball sampling technique. Foreign students, severely ill students, and house officers were excluded.

The sample size was calculated using NCSS PASS 11.0 and based on a study conducted by Ghonaim et al. [11]. Group sample sizes of approximately 93 in Group One (previously infected group) and 93 in Group Two (no previous infection) could achieve 80% power to detect a difference between group proportions of −0.1610. The proportions of Group One (treatment group) are assumed to be 0.8830 and 0.7220 under the null and alternative hypotheses, respectively. Conversely, the proportion of Group Two (control group) is 0.8830. The test statistics used is the two-sided $Z$-test with pooled variance. The significance level of the test was set to 0.0500. By the end of the study period, we recruited a total sample of 404 undergraduate medical students.

## Ethical considerations

Administrative approvals were obtained. An online consent was obtained from study participants which described the main study objectives and that the participation in the survey was voluntary, and withdrawal could be done whenever the participant wanted. Participants were informed that they can proceed with the survey if they select "I agree to participate in the study". The questionnaire was anonymous; data confidentiality was maintained. Documentation of informed consent was waived by the research ethics committee as the research data will be collected through a simple online survey. The study was approved by the research ethics committee, faculty of medicine, Ain Shams University (Approval number: FMASU R 201/ 2021). The study conformed to the international ethical guidelines and that of the Declaration of Helsinki (2013).

**Study tools.** A web-based self-administered anonymous questionnaire composed of 53 items was constructed on Google Form to collect data via a social media network (WhatsApp). Erfani et al. [12] designed the questionnaire on the basis of the WHO training material for the detection, prevention, response, and control of COVID-19 [13]. The authors of the questionnaire tested for validity and reliability, where five qualified experts, which included an infectious disease medical specialist, a physician–epidemiologist, an infection control nurse, a qualified general physician, and a community medicine specialist, reviewed content validity using the index of item objective congruence (IOC; 0.60–1). After confirming content and face validity, a pilot study using the instrument was conducted on a sample group of 30 participants. Cronbach's alpha of the questionnaire was calculated, which displayed good internal consistency ($\alpha = 0.73$). Prior to the completion of the final survey, the questionnaire was modified as necessary to promote a better understanding of the questions among the participants. Moreover, the study considered the arrangement of the questions to ensure its efficiency. Completing the questionnaire lasted for approximately 10–15 min. Specifically, the questionnaire consisted of five sections as follows. Section 1 focuses on demographic information (e.g., age, sex, residence, and academic year), whereas Section 2 inquires about previous COVID-19 infection status and how the infection was confirmed. Section 3 presents knowledge about COVID-19. This section consisted of 26 questions, which are designated as follows: six regarding the characteristics of the disease (K1–K6), six on the symptoms of the disease (K7–K12), and six regarding the prevention and control of the disease (K13–K18). Eight questions were further added (K19–K26), four of which asked about the transmission route of the disease, and four about groups at potentially high risk. Responses to these questions could be in a multiple-choice or a true-or-false format or Do not Know. Correct answers took a value of 1, whereas

incorrect or Do not Know responses took a value of 0. The overall knowledge score ranged from 0 to 26. Section 4 focuses on attitude toward COVID-19, which consisted of 15 questions (A1–A15). Responses were rated using a three-point Likert-type scale (1 = disagree, 2 = neutral, and 3 = agree). The total scores for attitude ranged from 15 to 45. Lastly, Section 5 presents questions on the practice of precautionary measures against COVID 19 by the medical students. This section was adopted from the recommended practices of the WHO for the prevention of the transmission of COVID-19. Responses were assessed using Yes or No questions related to hand washing, avoiding crowded places, maintaining social distancing (1 m), avoiding touching the face, and avoiding handshakes. The total number of items was 12 (P1–P12) with the following scoring system: correct responses took a value of 1; incorrect or Do not Know responses took a value of 0. The total scores ranged from 0 to 12.

## Statistical analysis

Data were revised for completeness and accuracy, coded, entered into a personal computer, and, finally, analyzed using SPSS version 20 (International Business Machine). Categorical data were presented as frequencies and related percentages, whereas quantitative data were presented as mean and standard deviation. The chi-squared test was used to examine the relationship between infection status, frequency of infection, and KAP levels. Fisher's exact test was used when 20% of the cells or more indicated an expected count of less than five. In addition, odds ratios and 95% confidence intervals (CIs) were calculated. Pearson's correlation coefficient was used to explore the associations between two qualitative variables.

## Results

The majority of medical students were female (64.1%) with ages ranging from 20 to 22 years (65.1%) and a mean age of 21.19 ± 1.75 years. The majority of the students were in their third and fourth academic years, which represent 33.2% and 21.0% of the sample, respectively. Moreover, 84.2% were urban residents. Regarding infection status of COVID 19; more than one-third (37.4%) were previously infected with COVID-19. A total of 60.5% mentioned that they were diagnosed through the signs and symptoms of COVID 19, whereas the others were diagnosed through PCR (nasopharyngeal swab; 17.1%) and preliminary laboratory testing (CBC/CRP/ferritin/D-dimer; 14.5%). In terms of the frequency of infection, 88.8% of the medical students mentioned that they were infected only once; 9.2% were infected twice; and only 2.0% were infected three times or more. For the frequency of smoking and chronic diseases among the students, 4.2% were smokers, and 6.2% reported chronic diseases (Table 1).

The study noted a statistically significant relationship between academic year and infection status, where 47.0%, 54.2%, and 43.3% of the medical students in their third, fifth, and sixth academic years, respectively, reported that they were previously infected with COVID-19 ($P < 0.001$). In addition, a statistically significant relationship was observed between the levels of knowledge and practice and infection status, where 45.8% and 46.3% of medical students with high levels of knowledge and practice reported a previous infection with COVID-19 ($P < 0.05$; Table 2).

Moreover, the study identified a statistically significant relationship between smoking status and frequency of COVID-19 infection; where ex-smokers (50.0%) and current smokers (16.7%) exhibited higher frequencies of infection than did non-smokers (9.9%; $P = 0.05$). In addition, a highly statistically significant relationship was found between levels of attitude and frequency of infection among medical students, where students with low levels of attitude (31.4%) reported a higher frequency of infection than did other students ($P < 0.01$). Moreover, medical students with low levels of knowledge and practice levels (22.2% and 16.7%,

respectively) exhibited a higher frequency of infection than did other students. However, this relationship did not reach a statistically significance level ($P > 0.05$; Table 3).

A statistically significant relationship was observed between the frequency of infection among medical students and their conformance to the practice of precautionary measures against COVID-19, such as mask wearing, maintaining social distancing in lecture halls, and frequent hand washing (33.3% and 66.7%, respectively). Those who reported that they did not conform to these practices experienced infection more than once ($P < 0.05$). Moreover, medical students with low levels of practice exhibited higher frequencies of infection compared with those of other students (16.7%). However, this relationship did not reach a statistically significant level ($P > 0.05$; Table 4).

A statistically significant negative correlation was observed between the number of previous infections among the students and their scores for knowledge and attitude toward COVID-19 ($P < 0.05$). In addition, a statistically significant positive correlation was identified among KAP scores ($P < 0.01$; Table 5).

The present study found that academic year and the presence of chronic diseases are statistically significant positive and negative predictors of COVID-19 infection status, respectively, among the medical students ($P < 0.05$; Table 6).

**Table 1. Characteristics of participants.**

| Variables | | No. | % |
|---|---|---|---|
| **Age** | Less than 20 years | 58 | 14.4% |
| | 20–22 years | 263 | 65.1% |
| | More than 22 years | 83 | 20.5% |
| | | **Mean ± SD** | **Range** |
| | | 21.19 ± 1.75 | (18.0–25.0) |
| **Gender** | Male | 145 | 35.9% |
| | Female | 259 | 64.1% |
| **Academic Year** | First year | 30 | 7.4% |
| | Second Year | 71 | 17.6% |
| | Third Year | 134 | 33.2% |
| | Fourth Year | 85 | 21.0% |
| | 5th Year | 24 | 5.9% |
| | 6th Year | 60 | 14.9% |
| **Residence** | Urban | 340 | 84.2% |
| | Rural | 64 | 15.8% |
| **Infection Status** | No | 253 | 62.6% |
| | Yes | 151 | 37.4% |
| **Are you a smoker?** | Non-Smoker | 382 | 94.6% |
| | Ex-Smoker | 5 | 1.2% |
| | Current Smoker | 17 | 4.2% |
| **Chronic Disease** | No | 379 | 93.8% |
| | Yes | 25 | 6.2% |
| **No. of Previous Infections (n = 151)** | Once | 135 | 88.8% |
| | Twice | 14 | 9.2% |
| | Three times or more | 3 | 2.0% |
| **Diagnosis (n = 151)** | Signs & Symptoms of COVID-19 | 92 | 60.5% |
| | Preliminary lab test (CBC/CRP/Ferritin/ᴅ-Dimer) | 22 | 14.5% |
| | Blood test (Antibodies) | 4 | 2.6% |
| | CT Scan | 8 | 5.3% |
| | PCR (Nasopharyngeal Swab) | 26 | 17.1% |

**Table 2. Relationship of infection status of medical students with their characteristics and levels of knowledge, attitude, and practice.**

| Variables | | Infection Status | | | | Odds Ratio (95% CI) | Chi-square test | P-value |
|---|---|---|---|---|---|---|---|---|
| | | No | | Yes | | | | |
| | | No. | % | No. | % | | | |
| Age | <20 years | 43 | 74.1% | 15 | 25.9% | | 4.608 | 0.100 |
| | 20–22 years | 163 | 62.0% | 100 | 38.0% | | | |
| | 22 years | 47 | 56.6% | 36 | 43.4% | | | |
| Gender | Male | 94 | 64.8% | 51 | 35.2% | 1.159 (0.760–1.769) | 0.469 | 0.493 |
| | Female | 159 | 61.4% | 100 | 38.6% | | | |
| Academic Year | First year | 27 | 90.0% | 3 | 10.0% | | 25.563 | 0.000** |
| | Second Year | 55 | 77.5% | 16 | 22.5% | | | |
| | Third Year | 71 | 53.0% | 63 | 47.0% | | | |
| | Fourth Year | 55 | 64.7% | 30 | 35.3% | | | |
| | Fifth Year | 11 | 45.8% | 13 | 54.2% | | | |
| | 6th Year | 34 | 56.7% | 26 | 43.3% | | | |
| Residence | Urban | 212 | 62.4% | 128 | 37.6% | 0.929 (0.533–1.620) | 0.067 | 0.795 |
| | Rural | 41 | 64.1% | 23 | 35.9% | | | |
| Smoking Status | Non-Smoker | 241 | 63.1% | 141 | 36.9% | | 3.666 (FE)# | 0.156 |
| | Ex-Smoker | 1 | 20.0% | 4 | 80.0% | | | |
| | Current Smoker | 11 | 64.7% | 6 | 35.3% | | | |
| Chronic Disease | No | 234 | 61.7% | 145 | 38.3% | 0.510 (0.199–1.306) | 2.037 | 0.154 |
| | Yes | 19 | 76.0% | 6 | 24.0% | | | |
| Knowledge Level | Low | 73 | 73.7% | 26 | 26.3% | | 7.128 | 0.028* |
| | Moderate | 167 | 59.4% | 114 | 40.6% | | | |
| | High | 13 | 54.2% | 11 | 45.8% | | | |
| Attitude level | Low | 60 | 63.8% | 34 | 36.2% | | 5.479 | 0.065 |
| | Moderate | 137 | 66.8% | 68 | 33.2% | | | |
| | High | 56 | 53.3% | 49 | 46.7% | | | |
| Practice Level | Low | 57 | 70.4% | 24 | 29.6% | | 7.522 | 0.023* |
| | Moderate | 123 | 65.8% | 64 | 34.2% | | | |
| | High | 73 | 53.7% | 63 | 46.3% | | | |

*Statistically significant at $P < 0.05$.

**Highly statistically significant at $P < 0.01$.

#Fisher's exact test was used when 20.0% of the cells or more exhibited an expected count of less than five.

Logistic regression analysis also indicated that the level of attitude is a statistically significant negative predictor of COVID-19 infection status among the medical students ($P < 0.05$; Table 7).

## Discussion

The present study revealed that more than one-third of the medical students (37.4%) reported being previously infected with COVID-19. The majority (60.5%) mentioned that they were diagnosed through the signs and symptoms of COVID-19 instead of PCR (17.1%). These findings are in accordance with those of Amjad et al. [14] who conducted a study on 1,830 medical students at the University of Jordan and found that 13% of the participants reported testing positive for COVID-19 infection through PCR.

This finding indicates that the numbers of positive cases through PCR that are officially announced do not reflect the true numbers of cases in the community, which may lead to the

**Table 3. Relationship of frequency of infection of medical students with their characteristics and levels of knowledge, attitude, and practice (n = 151).**

| Variables | | Frequency of infection | | | | Odds Ratio (95% CI) | Chi-square test | P-value |
|---|---|---|---|---|---|---|---|---|
| | | Once | | More than once | | | | |
| | | No. | % | No. | % | | | |
| **Age** | <20 years | 14 | 93.3% | 1 | 6.7% | | 0.509 | 0.851 |
| | 20–22 years | 90 | 89.1% | 11 | 10.9% | | | |
| | >22 years | 31 | 86.1% | 5 | 13.9% | | | |
| **Gender** | Male | 42 | 82.4% | 9 | 17.6% | 0.401 (0.145–1.113) | 3.227 | 0.072 |
| | Female | 93 | 92.1% | 8 | 7.9% | | | |
| **Academic Year** | First year | 4 | 100.0% | 0 | 0.0% | | 4.813 FE# | 0.383 |
| | Second Year | 15 | 93.8% | 1 | 6.2% | | | |
| | Third Year | 58 | 92.1% | 5 | 7.9% | | | |
| | Fourth Year | 23 | 76.7% | 7 | 23.3% | | | |
| | Fifth Year | 12 | 92.3% | 1 | 7.7% | | | |
| | Sixth Year | 23 | 88.5% | 3 | 11.5% | | | |
| **Residence** | Urban | 114 | 88.4% | 15 | 11.6% | 0.724 (0.154–3.400) | 0.179 FE# | 1.000 |
| | Rural | 21 | 91.3% | 2 | 8.7% | | | |
| **Smoking Status** | Non-Smoker | 128 | 90.1% | 14 | 9.9% | | 5.553 FE# | 0.050* |
| | Ex-Smoker | 2 | 50.0% | 2 | 50.0% | | | |
| | Current Smoker | 5 | 83.3% | 1 | 16.7% | | | |
| **Chronic Disease** | No | 130 | 89.0% | 16 | 11.0% | 1.625 (0.178–14.797) | 0.189 FE# | 0.515 |
| | Yes | 5 | 83.3% | 1 | 16.7% | | | |
| **Knowledge Level** | Low | 21 | 77.8% | 6 | 22.2% | | 3.786 FE# | 0.139 |
| | Moderate | 104 | 91.2% | 10 | 8.8% | | | |
| | High | 10 | 90.9% | 1 | 9.1% | | | |
| **Attitude level** | Low | 24 | 68.6% | 11 | 31.4% | | 20.947 | 0.000** |
| | Moderate | 67 | 98.5% | 1 | 1.5% | | | |
| | High | 44 | 89.8% | 5 | 10.2% | | | |
| **Practice Level** | Low | 20 | 83.3% | 4 | 16.7% | | 0.941 | 0.625 |
| | Moderate | 57 | 89.1% | 7 | 10.9% | | | |
| | High | 58 | 90.6% | 6 | 9.4% | | | |

*Statistically significant at $P < 0.05$.

**Highly statistically significant at $P < 0.01$.

#Fisher's exact test was used, when 20.0% of the cells or more displayed an expected count of less than five.

underestimation of the true situation of the disease. The reason for this notion is that the majority of the students exhibiting the signs and symptoms of COVID-19 isolate themselves at home and take medication without PCR testing or being recorded officially. PCR is an expensive test that is neither 100% sensitive nor specific; as such, students would rather save money spent on testing PCR to buy medications for treating their infection.

The findings of the current study highlighted that the majority of medical students were infected once; however, 11.2% were infected twice, thrice, or even more. This result indicates that medical students are highly exposed to COVID-19 infection in hospitals during clinical rounds, crowded lecture halls, social event gatherings, crowded transportation modes, or even their homes. These findings are in agreement with Barqawi et al. [15] who mentioned that physicians and medical students are the primary sectors in contact with patients with COVID-19, because they are participating in providing care services; thus, they are ultimately expected to be at higher risk to acquire the infection in comparison to the normal population.

**Table 4. Relationship between frequency of infection among medical students and practice related to precautionary measures against COVID-19 (n = 151).**

| | Items<br>To prevent contracting and spreading COVID-19 . . . | | No. of Previous Infections | | | | Odds Ratio (95% CI) | Chi-square test | P-value |
|---|---|---|---|---|---|---|---|---|---|
| | | | Once | | More than once | | | | |
| | | | No. | % | No. | % | | | |
| P1 | I avoid going out to crowded places. | Incorrect | 4 | 80.0% | 1 | 20.0% | 0.489 (0.051–4.644) | 0.405 FE# | 0.452 |
| | | Correct | 131 | 89.1% | 16 | 10.9% | | | |
| P2 | To prevent contracting and spreading COVID-19, I avoid unnecessary gatherings. | Incorrect | 17 | 89.5% | 2 | 10.5% | 1.081 (0.227–5.145) | 0.009 FE# | 1.000 |
| | | Correct | 118 | 88.7% | 15 | 11.3% | | | |
| P3 | To prevent contracting and spreading COVID-19, I avoid indoor activities. | Incorrect | 56 | 84.8% | 10 | 15.2% | 0.496 (0.178–1.383) | 1.848 FE# | 0.201 |
| | | Correct | 79 | 91.9% | 7 | 8.1% | | | |
| P4 | To prevent contracting and spreading COVID-19, I avoid handshaking, hugging, and kissing. | Incorrect | 16 | 84.2% | 3 | 15.8% | 0.627 (0.162–2.425) | 0.464 FE# | 0.449 |
| | | Correct | 119 | 89.5% | 14 | 10.5% | | | |
| P5 | To prevent contracting and spreading COVID-19, I avoid public transportations (taxi, bus, subway, plane, and train). | Incorrect | 41 | 83.7% | 8 | 16.3% | 0.491 (0.177–1.362) | 1.925 FE# | 0.178 |
| | | Correct | 94 | 91.3% | 9 | 8.7% | | | |
| P6 | To prevent contracting and spreading COVID-19, I wear masks and practice social distancing in lecture halls. | Incorrect | 8 | 66.7% | 4 | 33.3% | 0.205 (0.054–0.773) | 6.434 FE# | 0.031* |
| | | Correct | 127 | 90.7% | 13 | 9.3% | | | |
| P7 | To prevent contracting and spreading COVID-19, I frequently wash my hands. | Incorrect | 1 | 33.3% | 2 | 66.7% | 0.056 (0.005–0.655) | 9.484 FE# | 0.033* |
| | | Correct | 134 | 89.9% | 15 | 10.1% | | | |
| P8 | To prevent contracting and spreading COVID-19, I pay more attention to my personal hygiene than usual. | Incorrect | 5 | 83.3% | 1 | 16.7% | 0.615 (0.068–5.604) | 0.189 FE# | 0.515 |
| | | Correct | 130 | 89.0% | 16 | 11.0% | | | |
| P9 | To prevent contracting and spreading COVID-19, I use disinfectants and solutions. | Incorrect | 8 | 80.0% | 2 | 20.0% | 0.472 (0.092–2.434) | 0.837 FE# | 0.310 |
| | | Correct | 127 | 89.4% | 15 | 10.6% | | | |
| P10 | To prevent contracting COVID-19, I eat lots of fruits and vegetables. | Incorrect | 42 | 91.3% | 4 | 8.7% | 1.468 (0.452–4.769) | 0.411 FE# | 0.590 |
| | | Correct | 93 | 87.7% | 13 | 12.3% | | | |
| P11 | To prevent contracting COVID-19, I take vitamin C supplements. | Incorrect | 51 | 91.1% | 5 | 8.9% | 1.457 (0.485–4.376) | 0.454 FE# | 0.600 |
| | | Correct | 84 | 87.5% | 12 | 12.5% | | | |
| P12 | To prevent contracting and spreading COVID-19, face masks should be used in public and crowded places. | Incorrect | 92 | 89.3% | 11 | 10.7% | 1.167 (0.405–3.364) | 0.082 FE# | 0.787 |
| | | Correct | 43 | 87.8% | 6 | 12.2% | | | |
| | Practice level | Low | 20 | 83.3% | 4 | 16.7% | | 1.113 FE# | 0.651 |
| | | Moderate | 57 | 89.1% | 7 | 10.9% | | | |
| | | High | 58 | 90.6% | 6 | 9.4% | | | |

*Statistically significant at P < 0.05.

#Fisher's exact test was used when 20.0% of the cells or more display an expected count of less than five.

The present study revealed that students in their clinical years are significantly more exposed to infection instead of those during their preclinical years; This finding is in accordance with that of Amjad et al. [14] who found that a higher percentage of clinical students compared with preclinical students tested positive for COVID-19 (15.2% vs. 11.2%). The authors explained that this difference may be attributed to the study or work environment, where clinical students are more prone to encounter patients with COVID-19.

This study demonstrated that 45.8% and 46.3% of medical students with high levels of knowledge and practice, respectively, reported previous COVID-19 infection. This surprising finding could be explained by the presence of a confounding factor, that is, academic year, because students who are in the later academic years possess higher levels of knowledge and practice than those of other students. In addition, the infection rate is higher during clinical years instead of the preclinical year. This finding was further supported by the results of

**Table 5. Correlation between scores for frequency of infections, knowledge, attitude, and practice.**

| | | No. of Previous Infections | Knowledge Score | Attitude Score | Practice Score |
|---|---|---|---|---|---|
| **No. of Previous Infections** | Pearson's Correlation | 1 | −.202 | −.187 | −.089 |
| | Sig. (2-tailed) | | .013* | .021* | .274 |
| **Knowledge Score** | Pearson's Correlation | −.202 | 1 | .263 | .184 |
| | Sig. (2-tailed) | .013* | | .000** | .000** |
| **Attitude Score** | Pearson's Correlation | −.187 | .263 | 1 | .234 |
| | Sig. (2-tailed) | .021* | .000** | | .000** |
| **Practice Score** | Pearson's Correlation | −.089 | .184 | .234 | 1 |
| | Sig. (2-tailed) | .274 | .000** | .000** | |

*Correlation is significant at the 0.05 level (2-tailed).

**Correlation is significant at the 0.01 level (2-tailed).

multiple linear regression analysis that intended to eliminate the aforementioned confounder, that led to the disappearance of this relationship.

The present study also revealed that ex-smokers and current smokers exhibited significantly higher frequencies of infection than non-smokers (50.0%, 16.7% versus 9.9%; $P = 0.05$). To date, scholars assume that smoking may be associated with adverse disease prognosis, because extensive evidence has highlighted the negative impact of tobacco use on lung health and its causal association with a plethora of respiratory diseases [16].

Moreover, medical students with low levels of KAP exhibited higher frequencies of COVID-19 infection compared with those of other students. The results of Pham et al. [17] supported this finding, where the authors stated that medical students should continue to promote knowledge and attitude, which play a vital role in increasing adherence to healthy practices as well as in passing acquired knowledge to family, friends, or relatives to help in the fight against this pandemic and to decrease infection rates.

The present study provided evidence of a statistically significant negative correlation between the number of previous COVID-19 infections of the medical students and their scores for knowledge and attitude toward COVID-19 ($P < 0.05$). Furthermore, the study found a statistically significant positive correlation among KAP scores ($P < 0.01$). These findings are in accordance with those of Al-Rawajfah et al. [18] who found a significant positive relationship

**Table 6. Logistic regression of independent predictors of COVID-19 infection status.**

| Variables | B | S.E. | Wald | Sig. | Exp(B) | 95% C.I. for EXP(B) | |
|---|---|---|---|---|---|---|---|
| | | | | | | Lower | Upper |
| **Age** | −.497 | .304 | 2.673 | .102 | .609 | .336 | 1.104 |
| **Gender** | .046 | .210 | .048 | .827 | 1.047 | .694 | 1.580 |
| **Academic year** | .335 | .134 | 6.273 | .012* | 1.397 | 1.075 | 1.816 |
| **Residence** | −.410 | .278 | 2.171 | .141 | .664 | .385 | 1.145 |
| **Smoker** | .017 | .253 | .004 | .947 | 1.017 | .620 | 1.669 |
| **Chronic Disease** | −1.376 | .485 | 8.037 | .005** | .253 | .098 | .654 |
| **Knowledge level** | .255 | .210 | 1.484 | .223 | 1.291 | .856 | 1.947 |
| **Attitude level** | .100 | .152 | .434 | .510 | 1.106 | .820 | 1.490 |
| **Practice level** | .239 | .150 | 2.546 | .111 | 1.270 | .947 | 1.704 |

*Statistically significant at $P < 0.05$.

**Highly statistically significant at $P < 0.01$.

**Table 7. Logistic regression of independent predictors of the frequency of COVID-19 infection.**

| Variables | B | S.E. | Wald | Sig. | Exp(B) | 95% CI for EXP(B) | |
|---|---|---|---|---|---|---|---|
| | | | | | | Lower | Upper |
| Age | .022 | .128 | .030 | .862 | 1.023 | .796 | 1.314 |
| Gender | −.968 | .605 | 2.558 | .110 | .380 | .116 | 1.244 |
| Academic year | .232 | .280 | .688 | .407 | 1.261 | .729 | 2.184 |
| Residence | .111 | .878 | .016 | .899 | 1.117 | .200 | 6.251 |
| Smoker | .313 | .539 | .337 | .562 | 1.367 | .475 | 3.936 |
| Chronic Disease | .870 | 1.289 | .455 | .500 | 2.386 | .191 | 29.867 |
| Knowledge level | −.759 | .544 | 1.946 | .163 | .468 | .161 | 1.360 |
| Attitude level | −.994 | .407 | 5.960 | .015* | .370 | .167 | .822 |
| Practice level | −.018 | .403 | .002 | .965 | .982 | .446 | 2.165 |

*Statistically significant at $P < 0.05$.

among the knowledge, attitude, and precautionary practices of students. Hence, the authors stressed the important role and responsibility of the educators of health professionals in improving their knowledge, which will definitely aid in strengthening their role in fighting the current pandemic.

The study conducted multivariate logistic regression analysis to identify the significant predictors of COVID-19 infection status and found that academic year and the presence of chronic diseases are statistically significant predictors among medical students ($P < 0.05$). The findings of Ruba et al. [19] provided support for the current findings, who found that students from higher years of study were more knowledgeable and more prone to work with patients with COVID-19 compared with those in their first years of study. Given the relationship between chronic disease and COVID-19 infection, Gimeno-Miguel et al. [20] mentioned that patients with more severe forms of infection are typically affected by more than one chronic disease, which may explain the nature of chronic diseases as a negative predictor of COVID-19 infection in the present study. Specifically, they cause severe forms of the disease in terms of morbidity and increase the rates of mortality from COVID-19, which renders students with chronic diseases to become very cautious about contracting the infection and, hence, to display low rates of infection.

In addition, attitude level was found to be a statistically significant negative predictor of COVID-19 infection status among the medical students ($P < 0.05$).

Students may be infected with COVID-19 once due to chance or deliberately by being in close contact with a diseased family member. However, being infected more than once is frequently influenced by certain risk factors, such as the lack of information, negative attitudes, and non-conformance to precautionary measures against COVID-19, which, in turn, contributes to the high rates and frequencies of infection.

Many studies linked the KAP of medical students toward precautionary measures related to COVID-19 to their sociodemographic characteristics and academic grade. The present study explored a new perspective, that is, the frequency of risk factors of COVID-19 infection among medical students. The current study intends to deliver a message to the general population and not only to medical students: acquiring a significant amount of information and holding a positive attitude toward COVID-19 will definitely positively influence their behaviors and, hence, decrease infection rates and frequencies in the community.

The present study followed a sound methodology that enables the generalization of results to the population from which the sample was derived by calculating adequate sample size and

by recruiting medical students from various academic years. However, this study has its limitations, which should be considered. First, the collected data were obtained through an online survey and limited to one governmental medical university only, which limits the generalizability of the study. In general, a cross-sectional study could only pinpoint an association between risk factors and outcomes; however, it could not establish a causal relationship.

## Conclusion and recommendation

The present study concluded that medical students who were ex-smokers and current smokers with low levels of KAP exhibited higher frequencies of COVID-19 infection compared with those of other students. Moreover, the study observed a statistically significant negative correlation between the number of previous COVID-19 infections among medical students and their scores for knowledge and attitude toward COVID-19. In addition, the study identified a statistically significant positive correlation among KAP scores.

This study recommends that promoting the knowledge of medical students regarding precautionary measures against COVID-19, such as mask-wearing, hand washing, and maintaining social distancing, are crucial in reducing the risk of infection and, hence, saving lives.

## Supporting information

**S1 Data.**
(XLSX)

## Author Contributions

**Conceptualization:** Ghada O. Wassif.

**Data curation:** Ghada O. Wassif.

**Formal analysis:** Ghada O. Wassif, Dina Ahmed Gamal El Din.

**Methodology:** Ghada O. Wassif, Dina Ahmed Gamal El Din.

**Supervision:** Ghada O. Wassif.

**Validation:** Ghada O. Wassif, Dina Ahmed Gamal El Din.

**Writing – original draft:** Ghada O. Wassif, Dina Ahmed Gamal El Din.

**Writing – review & editing:** Ghada O. Wassif, Dina Ahmed Gamal El Din.

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
