## [Decision Letter · Decision Letter 0]

7 Apr 2022

PONE-D-22-02994Relationship between Knowledge, Attitude, and Practice of COVID-19 precautionary measures and the Frequency of infection among medical students at an Egyptian University.PLOS ONE

Dear Dr. Wassif,

Thank you for submitting your manuscript to PLOS ONE. After careful consideration, we feel that it has merit but does not fully meet PLOS ONE’s publication criteria as it currently stands. Therefore, we invite you to submit a revised version of the manuscript that addresses the points raised during the review process.Please put efforts in English and presentations, too. 

We look forward to receiving your revised manuscript.

Kind regards,

Jianguo Wang, PhD

Academic Editor

PLOS ONE

Journal Requirements:

Reviewers' comments:

Reviewer's Responses to Questions

**Comments to the Author**

1. Is the manuscript technically sound, and do the data support the conclusions?

Reviewer #1: Yes

Reviewer #2: Partly

2. Has the statistical analysis been performed appropriately and rigorously? 

Reviewer #1: I Don't Know

Reviewer #2: Yes

3. Have the authors made all data underlying the findings in their manuscript fully available?

Reviewer #1: Yes

Reviewer #2: Yes

4. Is the manuscript presented in an intelligible fashion and written in standard English?

Reviewer #1: No

Reviewer #2: No

5. Review Comments to the Author

Reviewer #1: the paper need English editing to be more interesting to the readers, more focus on the aim , connection between reinfection with COVID-19 among medical students and infection control procedures in the hospital

Reviewer #2: 1. Manuscript should be sent for English proofreading.

2. Referencing style must be followed as per journal’s guidelines.

3. Author’s must strengthen the rational/problem statement of the study objectives specific to the importance of KAP precautionary measures and medical students, of course previous studies won’t be available as COVID19 is a new phenomenon but authors can justify it in other ways.

4. Objectives must be added clearly in the introduction section

5. How was the validity and reliability of the questionnaire/study tool was established?

6. Ethical approval should be written n more scientific way excluding minor details like dean’s and vice dean’s approval

7. I wonder, how qualitative data can be presented in numbers?

8. Discussion must include the generalisability (external validity) of the study results

6. PLOS authors have the option to publish the peer review history of their article (what does this mean?). If published, this will include your full peer review and any attached files.

Reviewer #1: No

Reviewer #2: No

---

## [Author Response · Author response to Decision Letter 0]

7 May 2022

Dear Respectable Reviewers,

I really appreciate your precious time and your valuable comments that we hope we could matched your expectations and we would made the modifications as desired.

Reviewer #1:

1- The paper needs English editing to be more interesting to the readers.

The article had been submitted for English language editing by Enago, the editing brand of Crimson Interactive Inc. under Copyediting/Language editing and they provided us with a certificate proving; that is made available with the submitted files that the manuscript has been edited for English language, grammar, punctuation, and spelling. They also checked the adherence to journal guidelines and made required changes. 

2- More focus on the aim, the connection between reinfection with COVID-19 among medical students and infection control procedures in the hospital.

The following sentence had been added at the end of the introduction clarifying the research question and study objective “Page 4 Line 84-91”.

“The present study intends to explore if a relationship exists between the KAP of medical students related to COVID-19 precautionary measures and the frequency of infection in an endeavor to provide evidence that the conformance of medical students to COVID-19 precautionary measures can substantially reduce the incidence of infection. This objective is in line with the notion that infection may occur once due to inevitable contact with an infected family member. However, frequent infection could be prevented by practicing precautionary measures, which could decrease the burden of COVID-19 on the health care system and reduce the cost of treatment. In addition, mortalities associated with COVID-19 could be reduced effectively”.

A table displaying the relationship between frequency of infection in medical students and Practice of COVID-19 Precautionary measures had been added on Page 13-15.

Reviewer #2: 

1- The manuscript should be sent for English proofreading.

The article had been submitted for English language editing by Enago, the editing brand of Crimson Interactive Inc. under Copyediting/Language editing and they provided us with a certificate proving; that is made available with the submitted files that the manuscript has been edited for English language, grammar, punctuation, and spelling. They also checked the adherence to journal guidelines and made required changes. 

2- Referencing style must be followed as per the journal’s guidelines.

References had been revised and updated per the journal’s guidelines by the author and by Enago scientific editing services.

3-Authors must strengthen the rational/problem statement of the study objectives specific to the importance of KAP precautionary measures and medical students, of course, previous studies won’t be available as COVID19 is a new phenomenon, but authors can justify it in other ways.

Rational/study justification had been added at the end of the introduction section and the modifications had been marked with track changes (Page 4 Line 84-91).

Our study rationale is that infection may occur once due to inevitable contact with an infected family member. However, the frequent infection could be prevented by practicing precautionary measures, which could decrease the burden of COVID-19 on the health care system and reduce the cost of treatment. In addition, mortalities associated with COVID-19 could be reduced effectively”.

4-Objectives must be added clearly in the introduction section.

The following sentence had been added at the end of the introduction clarifying the research question and study objective “Page 4 Line 84-91”.

“The present study intends to explore if a relationship exists between the KAP of medical students related to COVID-19 precautionary measures and the frequency of infection in an endeavor to provide evidence that the conformance of medical students to COVID-19 precautionary measures can substantially reduce the incidence of infection. 

5-How was the validity and reliability of the questionnaire/study tool established?

The questionnaire was designed by Erfani et al., 2020 based on WHO training material for the detection, prevention, response, and control of COVID-19. Validity and reliability were tested by questionnaire authors and details had been added to the study tools in the methodology section on Page 5, Lines 113-121.

6-Ethical approval should be written in a more scientific way excluding minor details like the dean’s and vice dean’s approval.

The ethical approval statement was modified, and minor details were deleted with track changes.

7-I wonder, how qualitative data can be presented in numbers?

The word had been corrected on Page 6 Line 158; Categorical data as nominal and ordinal variables are presented in numbers and their related frequencies.

8-Discussion must include the generalizability (external validity) of the study results

A statement clarifying the generalizability (external validity) of study results had been added at the end of the discussion in addition to the study limitations on Page 21, lines 345-351.

---

## [Decision Letter · Decision Letter 1]

30 Aug 2022

Relationship between Knowledge, Attitude, and Practice of COVID-19 precautionary measures and the Frequency of infection among medical students at an Egyptian University.

PONE-D-22-02994R1

Dear Dr. Wassif,

We’re pleased to inform you that your manuscript has been judged scientifically suitable for publication and will be formally accepted for publication once it meets all outstanding technical requirements.

Kind regards,

Jianguo Wang, PhD

Academic Editor

PLOS ONE

Additional Editor Comments (optional):

Reviewers' comments:

Reviewer's Responses to Questions

**Comments to the Author**

1. If the authors have adequately addressed your comments raised in a previous round of review and you feel that this manuscript is now acceptable for publication, you may indicate that here to bypass the “Comments to the Author” section, enter your conflict of interest statement in the “Confidential to Editor” section, and submit your "Accept" recommendation.

Reviewer #1: All comments have been addressed

Reviewer #2: (No Response)

Reviewer #3: All comments have been addressed

2. Is the manuscript technically sound, and do the data support the conclusions?

Reviewer #1: Yes

Reviewer #2: No

Reviewer #3: Yes

3. Has the statistical analysis been performed appropriately and rigorously? 

Reviewer #1: Yes

Reviewer #2: No

Reviewer #3: Yes

4. Have the authors made all data underlying the findings in their manuscript fully available?

Reviewer #1: Yes

Reviewer #2: No

Reviewer #3: Yes

5. Is the manuscript presented in an intelligible fashion and written in standard English?

Reviewer #1: Yes

Reviewer #2: Yes

Reviewer #3: Yes

6. Review Comments to the Author

Reviewer #1: Dear author/ My thanks to you for all this efforts in revising and raising the paper making it matching Plos one preferences.

Reviewer #2: Although authors claims that they have made changes as per reviewer's comment but still I can't find those changes. There are no line numbers given which makes it very difficult to locate the changes.

Scores of Knowledge, attitude and practice are given without showing what were the knowledge questions. The supplement files are not useful at all they are just raw data.

Ethical consideration was asked to revise but they are still the same, no track changes found.

Some tables have unnecessary statistics information which can be revised to make it more statistically sound.

Overall, the quality of the manuscript is not up to the standards of PLOSONE.

Reviewer #3: Relationship between Knowledge, Attitude, and Practice of COVID-19 precautionary

measures and the Frequency of infection among medical students at an Egyptian

University

Dear Authors

Good Day

I suggest to Accept

Good Luck with your paper

7. PLOS authors have the option to publish the peer review history of their article (what does this mean?). If published, this will include your full peer review and any attached files.

Reviewer #1: No

Reviewer #2: No

Reviewer #3: **Yes: **Mainul Haque

---

## [Editor Report · Acceptance letter]

9 Sep 2022

PONE-D-22-02994R1 

Relationship between Knowledge, Attitude, and Practice of COVID-19 precautionary measures and the Frequency of infection among medical students at an Egyptian University 

Dear Dr. Wassif:

I'm pleased to inform you that your manuscript has been deemed suitable for publication in PLOS ONE. Congratulations! Your manuscript is now with our production department. 

Kind regards, 

on behalf of

Dr. Jianguo Wang 

Academic Editor

PLOS ONE